# Study on Reference Range of Zinc, Copper and Copper/Zinc Ratio in Childbearing Women of China

**DOI:** 10.3390/nu13030946

**Published:** 2021-03-15

**Authors:** Huidi Zhang, Yang Cao, Qingqing Man, Yuqian Li, Jiaxi Lu, Lichen Yang

**Affiliations:** National Institute for Nutrition and Health, Chinese Center for Disease Control and Prevention, Key Laboratory of Trace Element Nutrition, National Health Commission of the People’s Republic of China, Beijing 100050, China; zhanghuidi1114@126.com (H.Z.); Yasmine0814@163.com (Y.C.); manqq@ninh.chinacdc.cn (Q.M.); liyq@ninh.chinacdc.cn (Y.L.); lujx@ninh.chinacdc.cn (J.L.)

**Keywords:** reference range, copper, zinc, childbearing women

## Abstract

*Background:* Copper and zinc are both essential elements in humans, that play various biological roles in body functions. Population-based reference values have not yet been established in China especially in childbearing women. The aim of this study is to establish a reference value of Zn, Cu and Cu/Zn ratios in childbearing women aged 18–44 from a representative population in China. *Method:* A total of 191 healthy childbearing women aged 18–44 years old were enrolled from the China Adult Chronic Disease and Nutrition Surveillance (2015) in this study with a series strict inclusion criteria. Basic biological indicators (weight, height, waist, blood pressure, high-density lipoprotein cholesterol, low-density lipoprotein, total cholesterol, triglyceride, fast glycose, HbA1c, blood pressure, uric acid) and elements levels in plasma and whole blood were collected. The 2.5th to 97.5th was used to represent the reference range of Cu, Zn and Cu/Zn ratio. *Results:* The reference range of Zn, Cu and Cu/Zn ratio in plasma were 70.46–177.53 µg/dL, 74.30–170.68 µg/dL and 0.54–1.68, respectively. The reference range of Zn, Cu and Cu/Zn ratios in whole blood were 402.49–738.05, 74.63–124.52 and 0.13–0.25 µg/dL, respectively. *Conclusion:* The reference range of Zn, Cu and Cu/Zn ratios in plasma and whole blood of healthy Chinese childbearing women could be used as an indicator to evaluate the status of element deficiency and overload.

## 1. Introduction

Zinc is an important trace element. As a catalyst, structural element and regulatory ion [1], it participates in many metabolic processes. Zinc deficiency would affect multiple body functions, such as impaired growth, increased risk of child morbidity and mortality, and preterm births [2]. Zn plays an important role in maintaining health status throughout the entire life.

Cu is also an essential micronutrient and it associates with a variety of biological reduction and oxidation (redox) processes as a transition metal [3]. Higher than normal copper concentrations are toxic and harmful to human health [4]. Identifying marginal deficiency of Cu is somewhat problematic [5] and it needs to be controlled very closely.

The study of Sugiura [6] and Jaiser [7] have shown that Zn intake could cause Cu deficiency, as Zn and Cu may compete for binding to metallothionein [8] and intestinal uptake [9]. As a result, the ratio of copper to zinc (Cu/Zn) is very appropriate to represent the reserves and metabolic conditions of Zn and Cu in the body.

To explore the reference range of these metal elements in healthy people is not only helpful for the determination of exposure limits of public health departments, but also of great significance for the prevention and treatment of various diseases caused by the lack or excess of metal elements [10]. The balanced intake and absorption of these metals can make the body have normal metabolic activities and keep healthy.

China has not yet established a population-based reference value, especially for women of childbearing age. The aim of this study is to establish a reference value of Zn, Cu and Cu/Zn ratio in childbearing women aged 18–44 from a representative population in China.

## 2. Materials and Methods

### 2.1. Subjects

The study was obtained from the China Adult Chronic Disease and Nutrition Surveillance (2015), which is a national representative cross-sectional survey. The International Federation of Clinical Chemistry (IFCC) recommends at least 120 observations to estimate reliable reference values [11]. We selected healthy childbearing women if they were 18–44 with the following selection criteria. Inclusion criteria: (1) BMI: 18.5~24.0 kg/m^2^; (2) total cholesterol (TC) <5.2 mmol/L; (2) triglyceride (TG) <1.7 mmol/L; (4) low-density lipoprotein(LDL) <3.12 mmol/L; (5) high density lipoprotein cholesterol (HDL-C) >1.04 mmol/L; (6) uric acid (UA) ≤360 umol/L; (7) systolic blood pressure (SBP): 90–140 mmHg; (8) diastolic blood pressure (DBP): 60–89 mmHg; (9) fasting glucose (FG): 3.9–6.1 mmol/L; (10) hemoglobin A1c (HbA1c): 4–6%; (11) Hb: 115–150 g/L; (12) heart rate: 60–100 t/min. Exclusion criteria were (1) smoking or have a history of smoking; (2) drinking or have a history of drinking in the past 12 months. All subjects signed informed consent. This study was carried out in accordance with the declaration of Helsinki, and the project was authorized by the ethics committee of National Institute of nutrition and health, China CDC (file No.: 201519-a).

### 2.2. Data Collection

The physical examination was conducted by trained medical staff in accordance with standard procedures. Height is measured by a height meter with a minimum scale of 0.1 cm; the body weight is measured by electronic scale with a minimum scale of 0.1 kg. Body mass index (BMI) was calculated as weight (kg)/square of height (m^2^). Waist circumference is measured by a waist ruler with a length of 1.5 m, a width of 1 cm and a minimum scale of 0.1 cm. SBP (mmHg) and DBP (mmHg) were measured with Omron HBP1300 sphygmomanometer with an accuracy of 1 mmHg. After fasting for 10–14 h, 4 mL venous blood was collected and placed in the vacuum EDTA-K2 anticoagulant blood collection vessel. Remove 0.5 mL into the whole blood vessel. After 20–30 min at room temperature, the remaining blood sample was centrifuged at 1500× g for 15 min to obtain plasma sample. All blood samples were frozen at −80 °C for subsequent analysis and detection. Serum fasting glucose, HDL-C, LDL, TC, TG, UA were measured by an enzymatic method using automatic biochemical analyzer (Hitachi 7600, Tokyo, Japan). Hb was detected by the cyanide high iron method and HbA1c was tested by high performance liquid chromatography (HPLC, Waters e2695, Milford, MA, USA). The plasma and whole blood copper and zinc concentrations were measured by inductively coupled plasma mass spectrometry (ICP-MS, PerkinElmer, NexION 350, Waltham, MA, USA) from the anticoagulation tube.

### 2.3. Determination of Cu and Zn in the Plasma and Whole Blood

Plasma and whole blood samples were diluted with 0.5% (v/v) HNO3 (1:20) and 0.5% (v/v) HNO3 and 0.05% (v/v) Triton-X-100 (1:25), respectively. The concentrations of zinc and copper were analyzed by ICP-MS. Commercial quality control samples (Clincheck Level-2, Munich, Germany; Seronorm, Level-2, Billingstad, Norway) were used to monitor and analyze at intervals of 20 samples to ensure the precision and accuracy of detection. The recovery of Cu was 100.10% and 99.06% in plasma and whole blood, respectively. The recovery of Zn was 99.50% and 99.99% in plasma and whole blood, respectively. For plasma and whole blood, the inter-day precision of Zn for both were 1.94%, while the intra-day precision values were 2.71% and 3.74%. For plasma and whole blood, the inter-day precision of Cu for both were 1.31% and 1.71%, while the intra-day precision values were 1.53% and 3.96%. The Cu/Zn ratio was calculated by the concentration of Cu diving the concentration of Zn. As the unit of Cu and Zn were both ug/dL, we firstly need to transit the unit to umol/L. The concentrations multiply by 0.153 to get umol/L for Zn and multiply by 0.157 to get umol/L for Cu. In that case, we could get a molar ratio of Cu/Zn.

### 2.4. Statistic Analysis

SPSS software version 19.0 was used for statistical analysis. The results of descriptive characteristics were expressed as geometric mean (GM), median, P2.5 and P97.5. IFCC recommends that the reference range of clinical biomarkers be determined according to the concentration estimates of 2.5th and 97.5th in the reference population. The difference between two groups was detected with student t test. The difference between more than 2 groups was detected with one way ANOVA.

## 3. Results

### 3.1. Characteristics of the Study Population

This study was conducted on 191 childbearing women, aged 18–44 years old for determining serum and whole blood zinc, copper and Cu/Zn ratio reference values. The basic demographic and clinical data were analyzed for all these healthy individuals who met the inclusion and exclusion criteria. The characteristics of the participants are presented in Table 1.

### 3.2. Plasma and Whole Blood Concentrations of Zn in Healthy Childbearing Women

Table 2 shows the plasma and whole blood concentrations of Zn compared by age and area factors. The reference range of Zn in plasma was 70.46–177.53 µg/dL and in whole blood was 402.49–738.05 µg/dL. The concentrations of Zn in plasma has no significant difference in different age groups, area groups and residences. However, the whole blood Zn value showed significant difference between east area, west area and mid area.

### 3.3. Plasma and Whole Blood Concentrations of Cu in Healthy Childbearing Women

The concentrations of Cu in the plasma and whole blood of the 191 healthy childbearing women are shown in Table 3. The reference range of Cu in plasma was 74.30–170.68 µg/dL and in whole blood was 74.63–124.52 µg/dL. Plasma copper concentration was negatively associated with the geographical distribution from east to west (*p* = 0.015). Further analysis found that the difference between east area and mid area was statistically significant. A significant decreased of plasma copper concentration was observed in city residents compared with the rural area residents (*p* < 0.018). A significant difference in the whole blood copper concentration was also found among three area groups (*p* = 0.033). Whole blood copper level was significantly lower in the mid area compared with the east area.

### 3.4. Plasma and Whole Blood Cu:Zn Ratio in Healthy Childbearing Women

The reference ranges of Cu/Zn were 0.54–1.68 and 0.13–0.25 in plasma and whole blood, respectively. The Cu/Zn ratios were significantly different in the three areas in both plasma and whole blood. In whole blood, the Cu/Zn was significantly higher in the mid area than in the west area. However, in the plasma there was an inverse trend. The Cu/Zn was statistically higher in west area than in mid area. The information is shown in Table 4.

### 3.5. Plasma and Whole Blood Levels of Cu, Zn and Cu/Zn in Various Countries

The comparison of plasma and whole blood Zn, Cu and Cu/Zn levels with various countries are shown in Table 5.

## 4. Discussion

In this study we enrolled 191 healthy childbearing women from a national representative population with a series of strict screening criteria. Then, using whole blood and plasma samples, we measured copper and zinc, which are ubiquitous in food, water and the environment. The reference range of Cu, Zn and Cu/Zn ratio are defined with 2.5th to 97.5th, which is commonly used to represent a reference range now [23]. It could indicate the baseline exposure of healthy Chinese childbearing women aged 18 to 44 years old and may be used to identify increased exposure in an individual. At the same time, we also considered the influence of geographical location and age factors and incorporated these factors into the study. The new reference ranges for Cu, Zn and Cu/Zn differed by changing the subgroups of age groups, area and residence were also calculated.

From the first proof of the important role of zinc in microbiological system by Raulin [24] in 1896 to the Institute of Medicine first included zinc in RDAs [25] in 1974, and in 2004, the International Zinc Nutrition Consultative Group (IZiNCG) conducted a comprehensive review and evaluation of various aspects of zinc nutrition [26,27], the important role of Zn in biological function and the metabolism are gradually well known. However, due to the lack of definite reference value of zinc concentration, it is difficult to accurately evaluate the status of zinc. As mentioned by Apostoli [28], the reference range of metals should be established and updated regularly, because they may be affected by different environments due to different regions and gender. Additionally, in the cut-off detection of serum zinc concentrations in American NHANES [29], the assessment of zinc status was conducted by gender. The plasma and the whole blood reference interval of Zn in this study were 70.46–177.53 µg/dL and 402.49–738.65 µg/dL, respectively. The determination of plasma Zn is a good indicator of high or low Zn storage. The lower threshold of the reference values of our study was 70.46 µg/dL, which is consistent with the threshold 70 µg/dL recommended by the IZiNCG for judging zinc deficiency in adult women. Through the comparison of the data, we found that our results are at the same order of magnitude as Czech [12] (reference interval 58.79–112.93 µg/dL), Norway [13] (P2.5–P97.5 71.26–108.53 µg/dL) and Iran [14] (reference value 58.17–195.42 µg/dL). Compared with Pakistan [15] (reference rang 64.97–240.98 µg/dL), which belongs to the same Asian country, our reference value distribution range is narrow. And the whole blood Zn data here obtained (P2.5–P97.5 402.49–738.65 µg/dL) were fully comparable to those of Italy (407.6–759.4 µg/dL) [16], and slightly lower to Czech [17] (564.0–844.3 µg/dL). In a study of 20–45 years old healthy women in Wuhan, China [18], the results (P5–P95, 431.00–687.00 µg/dL) were similar to ours. Compared with the same population and type of study in different countries, the distribution range of plasma and whole blood was more consistent, even though there were differences in race, environment and economic level. The reference range of subgroup individuals is proposed because of the change of zinc in whole blood with regional distribution (east: 427.58–738.85 µg/dL; mid: 389.25–731.65 µg/dL; west: 430.26–750.89 µg/dL). The nutritional status of the population may vary due to differences in economic development in different regions.

Considering Cu, the level of Cu could reflect the recent exposure due to it is sequestered in blood by the ceruloplasmin [30]. The reference range of Cu in plasma was 74.30–170.68 µg/dL. This value was much higher than that for Korea [19] (67.8–154.5 µg/dL). Because area influence the level of Cu (*p* = 0.015), two different reference intervals were calculated for east area (75.87–182.85 µg/dL) and mid area (76.56–146.37 µg/dL). Additionally, the residence subgroup was also statistically associated with Cu values (*p* = 0.018). The whole blood reference range of Cu was (74.63–124.52 µg/dL). In a German population [20] Cu blood was found in the range 80.4–162.0 µg/dL, which is much higher than the range of our study. Sweden [21], the value range (69.0–147.5 µg/dL) was much boarder than us. In the study of healthy women in Wuhan, China [18], the reference range of Cu (62.95–104.11 µg/dL) was much lower than that was in our research. Additionally, in a study of the Chinese population [22] the lower level (71.9 µg/dL) was similar with us, but the upper level (211.3 µg/dL) was higher than that of us. Additionally, in the whole blood, the influence of area was found. The age of the subjects had no influence on blood Cu levels.

As mentioned by Van [30], considering the antagonistic effect of zinc and copper, when the ratio of zinc to copper in serum is close to 1:1, the immune response to the infectious agents is more effective. Zinc is in balance with copper in the blood, and a small increase in zinc consumption in excess of the recommended amount has been shown to reduce copper status [31,32]. The ratio of Cu/Zn not only reflects the individual level of trace elements, but also seems to have an important impact on metabolism, indicating that these trace elements play an important role in the pathogenesis of metabolic diseases [33,34]. The reference intervals of Cu/Zn ratio in this study were 0.54–1.68 in plasma and 0.13–0.25 in whole blood. The range was consistent with another study hosted in China (0.57–1.63) [22].

The primary strength of this study is its ability to provide national estimates of the reference value in 18–44 years childbearing population using a high degree of both inclusion criteria and laboratory quality control. Secondly, in this study we effort to establish reference range of metals in a variety of human media, including plasma and whole blood. The establishment of these reference values is conducive to maintaining the enzyme activity in the body and a healthy state.

There are some limitations in this study. Firstly, the dietary intakes were not available in this study. So, there are some weaknesses in evaluating the influence of diet on the reference range Secondly, it was difficult to estimate the influence of inflammation on elements nutritional status due to the lack of correction of inflammatory indicators.

## 5. Conclusions

In conclusion, this study presented the reference ranges of Cu, Zn and Cu/Zn in plasma and whole blood of 18–44 years childbearing women in China. The data can be used to assess the clinical health and body burden of this population.

## Figures and Tables

**Table 1 nutrients-13-00946-t001:** General characteristics of the healthy childbearing women.

Variables	Median	GM	P25	P75
Age (years)	27.98	28.60	24.05	34.43
Height (cm)	157.30	157.15	153.38	161.43
Weight (kg)	51.80	51.93	48.68	55.15
BMI (kg/m^2^)	21.18	21.03	19.85	22.26
Waist (cm)	71.40	71.46	67.59	75.29
TC (mmol/L)	4.11	4.05	3.73	4.48
TG (mmol/L)	0.71	0.73	0.58	0.96
LDL (mmol/L)	2.32	2.25	2.00	2.66
HDL (mmol/L)	1.40	1.40	1.24	1.54
UA (umol/L)	242.35	235.91	205.88	279.33
SBP (mmHg)	113.83	114.02	107.67	120.67
DBP (mmHg)	70.33	70.90	66.67	75.67
FG (mmol/L)	4.87	4.85	4.62	5.14
HbA1c (%)	4.80	4.76	4.48	5.20
Hb (g/L)	137.91	135.70	128.36	144.15
Heart Rate (t/min)	76.67	77.50	72.00	83.33

BMI, body mass index; TC, total cholesterol; TG, triglycerides; LDL, low-density lipoprotein; HDL-C, high density lipoprotein cholesterol; UA: uric acid; SBP, systolic blood pressure; DBP, diastolic blood pressure; FG, fasting glucose; HbA1c, hemoglobin A1c.

**Table 2 nutrients-13-00946-t002:** Plasma and whole blood zinc concentration in healthy childbearing women.

Variables	Plasma (µg/dL)						Whole Blood (µg/dL)				
	N	GM	P50	P2.5	P97.5	*p*	GM	P50	P2.5	P97.5	*p*
Total	191	110.32	112.13	70.46	177.53		539.29	540.75	402.49	738.05	
Age group						0.809					0.445
18–25 years	73	110.40	113.73	64.42	180.28		548.49	553.96	382.81	751.53	
26–35 years	72	111.11	109.19	70.54	196.37		531.29	517.93	386.79	744.28	
36–45 years	46	108.86	110.75	72.07	160.29		537.64	545.40	406.81	733.58	
Area						0.492					0.003
East	70	108.85	109.77	64.28	183.74		556.41	558.11	427.58	738.85	
Mid	91	109.77	112.92	71.21	173.32		519.04 *	515.29	389.25	731.65	
West	30	115.78	118.63	73.05	196.33		566.16 ^#^	590.13	430.56	750.89	
Residences						0.577					0.074
City	83	111.52	113.90	65.86	175.35		551.23	554.05	406.71	754.51	
Rural area	108	109.41	109.95	72.76	183.29		530.07	522.86	391.38	725.85	

* Compared with east, *p* < 0.05; ^#^ compared with mid, *p* < 0.05.

**Table 3 nutrients-13-00946-t003:** Plasma and whole blood copper concentration in healthy childbearing women.

Variables	Plasma (µg/dL)						Whole Blood (µg/dL)				
	N	GM	P50	P2.5	P97.5	*p*	GM	P50	P2.5	P97.5	*p*
Total	191	104.71	102.95	74.30	170.68		92.98	92.36	74.63	124.52	
Age group						0.791					0.336
18–25 years	73	104.06	102.86	66.27	161.39		93.89	93.77	66.63	129.56	
26–35 years	72	105.53	102.79	70.68	180.47		91.36	91.45	70.67	117.71	
36–45 years	46	104.36	104.87	76.11	164.31		94.09	91.25	77.68	134.78	
Area						0.015					0.033
East	70	110.92	107.7	75.87	182.85		96.07	95.42	77.52	126.3	
Mid	91	101.49 *	98.17	76.56	146.37		91.04 *	90.42	68.56	118.82	
West	30	100.71	96.79	46.84	169.34		91.87	91.61	48.32	136.76	
Residences						0.018					0.598
City	83	100.53	98.74	70.61	177.06		92.48	93.86	67.97	116.77	
Rural area	108	108.00	107.52	74.53	171.26		93.37	90.85	75.47	129.74	

* Compared with east, *p* < 0.05.

**Table 4 nutrients-13-00946-t004:** Plasma and whole blood Cu/Zn in healthy childbearing women.

Variables	Plasma						Whole Blood				
	*N*	GM	P50	P2.5	P97.5	*p*	GM	P50	P2.5	P97.5	*p*
Total	191	0.91	0.89	0.54	1.68		0.18	0.18	0.13	0.25	
Age group						0.836					0.593
18–25 years	73	0.90	0.84	0.51	1.77		0.18	0.18	0.12	0.25	
26–35 years	72	0.91	0.90	0.55	1.75		0.18	0.18	0.13	0.28	
36–45 years	46	0.93	0.92	0.38	1.73		0.17	0.17	0.09	0.32	
Area						0.037					0.020
East	70	0.96	0.93	0.48	1.92		0.17	0.17	0.13	0.28	
Mid	91	0.91	0.91	0.53	1.66		0.18	0.18	0.13	0.26	
West	30	0.81	0.77	0.55	1.91		0.17	0.17	0.13	0.21	
Residences						0.779					0.669
City	83	0.92	0.90	0.52	1.63		0.18	0.18	0.12	0.26	
Rural area	108	0.90	0.87	0.54	1.80		0.18	0.18	0.13	0.27	

^#^ Compared with mid, *p* < 0.05.

**Table 5 nutrients-13-00946-t005:** Plasma and whole blood levels of Cu, Zn and Cu/Zn concentration in various countries.

Element	Country	Parameter	Concentrations (µg/dL)
			Plasma	Whole Blood
Zn (µg/dL)	China (this study)	P2.5–P97.5	70.46–177.53	402.49–738.10
Czech [12]	reference interval	58.79–112.93	
Norway [13]	P2.5–P97.5	71.26–108.53	
Iran [14]	Reference value	58.17–195.42	
Pakistan [15]	Reference range	64.97–240.98	
Italy [16]	Reference range		407.60–759.40
Czech [17]	P25–P95		564.00–844.30
China [18]	P5–P95		431.00–687.00
Cu (µg/dL)	China(this study)	P2.5–P97.5	74.30–170.68	74.63–124.52
Korea [19]	P2.5–P97.5	67.80–154.50	
Germany [20]	P5–P95		80.40–162.00
Sweden [21]	min–max		69.00–147.50
China [18]	P5–P95		62.95–104.11
China [22]	P2.5–P97.5		71.90–211.30
Cu/Zn	China(this study)	P2.5–P97.5	0.53–1.65	0.12–0.25
China [22]	P2.5–P97.5	0.57–1.63

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
