# Peer review of "Study on Reference Range of Zinc, Copper and Copper/Zinc Ratio in Childbearing Women of China"

_nutrients, 2021, doi:10.3390/nu13030946_

Round 1

Reviewer 1 Report

First, the English grammar is an enormous issue.  Almost every sentence of the introduction on the first page has a major grammatical error.  I stop looking after this.

Next, why have the authors measured a whole series of characteristics and not looked for any relationships between them and the plasma Cu, Zn, and Cu/Zn?

In table 5, blank lines or some other divider is needed to separate entries for Zn, Cu, and Cu/Zn ratio.  I cannot clearly tell where one stops and another starts.

I find it difficult to believe that there are so few studies to compare results against.  I quick search of PubMed for copper, zinc, pregnancy, and China found a few.

See, for example,:

Biological Trace Element Research doi: 10.1007/s12011-020-02337-y

Ecotoxicology and Environmental Safety Vol, 210, 111854, 2021.

 Author Response

Dear professor,

We very appreciate your careful reading of our manuscript and the valuable suggestions. We have carefully considered the comments and revised the manuscript accordingly. The comments can be summarized as follows:

Point 1: First, the English grammar is an enormous issue.  Almost every sentence of the introduction on the first page has a major grammatical error.  I stop looking after this.

Response 1: Thank you so much for the kind suggestion. We are so sorry for the trouble caused by the inappropriate description. We will correct the grammatical errors in the article.

Point 2:Next, why have the authors measured a whole series of characteristics and not looked for any relationships between them and the plasma Cu, Zn, and Cu/Zn?

Response 2: Thank you so much for the kind suggestion. The aim of this study is to establish a reference value of Zn, Cu and Cu/Zn ratio in childbearing women aged 18-44 from a representative population in China. And we measured a whole series of characteristics in order to screen healthy people more comprehensively. So in this study we are not looked for any relationships between them and the plasma Cu, Zn, and Cu/Zn.

Point 3: In table 5, blank lines or some other divider is needed to separate entries for Zn, Cu, and Cu/Zn ratio.  I cannot clearly tell where one stops and another starts.

Response 3: Thank you so much for the kind suggestion. We've added borders to different entries to make it easier to distinguish the categories of different elements.

Point 4: I find it difficult to believe that there are so few studies to compare results against. I quick search of PubMed for copper, zinc, pregnancy, and China found a few.

See, for example,:

Biological Trace Element Research doi: 10.1007/s12011-020-02337-y

Ecotoxicology and Environmental Safety Vol, 210, 111854, 2021.

Response 4: Thank you so much for the kind suggestion. We are very appreciated about your help and studied these literature carefully. For the discussion part, we add some related literature and compare them with the results of this paper. The addition has been highlighted in the manuscript.

Reviewer 2 Report

You wrote that one of exclusion criteria was drinking: do you mean that the childbearing hadn't have drank alchol before? Please explain better.

You spoke about HPLC and ICP. Please insert model and brands.

Author Response

Dear professor,

We very appreciate your careful reading of our manuscript and the valuable suggestions. We have carefully considered the comments and revised the manuscript accordingly. The comments can be summarized as follows:

Point 1:You wrote that one of exclusion criteria was drinking: do you mean that the childbearing hadn't have drank alchol before? Please explain better.

Response 1: Thank you so much for the question. We determined the drinking status of the subjects from the questionnaire of China Adult Chronic Disease and Nutrition Surveillance according to whether they had drunk in the past 12 months. Individuals who did not drink alcohol in the past 12 months were included in the study. And we added the details of this in the manuscript.

Point 2: You spoke about HPLC and ICP. Please insert model and brands.

Response 2: Thank your for the kind suggestion. The model and brand of ICP-MS is NexION 350 from Perkin Elmer, America and of HPLC is Water e2695 from America. And we have already added it in the article.

Reviewer 3 Report

Εnglish language is adequate

Material and Methods are well structured and Results are well presented.

More correlations with other studies are needed in Discussion part.

Author Response

Dear professor,

We very appreciate your careful reading of our manuscript and the valuable suggestions. We have carefully considered the comments and revised the manuscript accordingly. The comments can be summarized as follows:

Point 1:More correlations with other studies are needed in Discussion part.

Response 1:Thank you so much for the kind suggestion. We have added some related literature and compare them with the results of this paper in the discussion part.

This manuscript is a resubmission of an earlier submission. The following is a list of the peer review reports and author responses from that submission.